# Reranking Laws for Language Generation:
# A Communication-Theoretic Perspective

**António Farinhas**[1,2]     **Haau-Sing Li**[2,3]     **André F. T. Martins**[1,2,4,5]
[1]Instituto Superior Técnico, Universidade de Lisboa     [2]Instituto de Telecomunicações
[3]Ubiquitous Knowledge Processing Lab, TU Darmstadt     [4]ELLIS Unit Lisbon     [5]Unbabel
{antonio.farinhas,andre.t.martins}@tecnico.ulisboa.pt, hli@ukp.tu-darmstadt.de

## Abstract

To ensure large language models (LLMs) are used safely, one must reduce their propensity to hallucinate or to generate unacceptable answers. A simple and often used strategy is to first let the LLM generate multiple hypotheses and then employ a reranker to choose the best one. In this paper, we draw a parallel between this strategy and the use of redundancy to decrease the error rate in noisy communication channels. We conceptualize the generator as a sender transmitting multiple descriptions of a message through parallel noisy channels. The receiver decodes the message by ranking the (potentially corrupted) descriptions and selecting the one found to be most reliable. We provide conditions under which this protocol is asymptotically error-free (*i.e.*, yields an acceptable answer almost surely) even in scenarios where the reranker is imperfect (governed by Mallows or Zipf-Mandelbrot models) and the channel distributions are statistically dependent. We use our framework to obtain reranking laws which we validate empirically on two real-world tasks using LLMs: text-to-code generation with DeepSeek-Coder 7B and machine translation of medical data with TowerInstruct 13B.

## 1 Introduction

Large language models (LLMs) have shown remarkable performance across many tasks in natural language processing, computer vision, and speech recognition. Despite their capabilities, instances of hallucinations and other critical errors occasionally arise, casting doubt on the reliability of their predictions, without clear indication of when and how badly they might fail (Ji et al., 2023; Guerreiro et al., 2023). This is particularly concerning as these models are increasingly used in high-stakes applications such as those within the medical or legal domains (Hung et al., 2023) or as agents that can perform multiple tasks, including generating and executing code (Wang et al., 2024).

The most common mitigation strategy is to "steer" the LLM with the aid of a reward model or directly from human preferences, either at training time (Stiennon et al., 2020; Yuan et al., 2024; Rafailov et al., 2024) or during decoding (Liu et al., 2024; Huang et al., 2024). A simple and effective decoding-time strategy is first to generate multiple hypotheses and then use a reranker to select the most appropriate one. Several generation techniques used with modern LLMs, including voting procedures (Borgeaud and Emerson, 2020; Wang et al., 2023; Liévin et al., 2024; Shi et al., 2022), minimum Bayes risk decoders (Eikema and Aziz, 2020; Freitag et al., 2022), quality-aware decoders (Fernandes et al., 2022), or other types of hypothesis ensembling/reranking techniques (Farinhas et al., 2023; Ni et al., 2023; Bertsch et al., 2023; Li et al., 2024), embody this idea. An essential aspect of these procedures is that they all add **redundancy** as an intermediate step (by generating multiple hypotheses) to increase the chances of returning an acceptable answer as the final output.

The idea of adding redundancy to decrease the error rate in noisy channels is a cornerstone of **communication theory**, more specifically in forward error correction methods. In its simplest

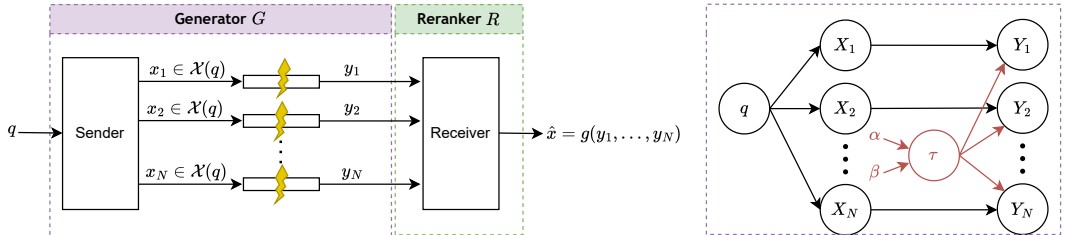

Figure 1: **Left:** A generator-reranker system $(G, R)$ depicted as a communication system (§2). Given a query $q$ with acceptance set $\mathcal{X}(q)$, the sender sends $N$ descriptions through noisy channels. The receiver's goal is to decode an acceptable answer through reranking. **Right:** Graphical model of the generator $G$. We consider two different models: a simplified version with $N$ independent hypotheses, represented in black (§3), and a scenario with exchangeable hypotheses, represented in red (§4).

form—repetition codes—a message block is sent multiple times, and the decoder uses some form of majority voting to recover the original message with high probability (MacKay, 2002; Cover and Thomas, 2006). The same idea underlies more sophisticated error-correcting codes (Hamming, 1950; Reed and Solomon, 1960; Gallager, 1962; Berrou et al., 1993).

In this paper, we draw a parallel between these two worlds by regarding generator-reranker LLMs as communication systems (§2 and Fig. 1, left). We conceptualize the LLM generator $G$ as a sender transmitting $N$ message descriptions in parallel through noisy channels, leading to $N$ potentially corrupted hypotheses. Then, the receiver, which corresponds to the reranker $R$, decodes the message by ranking the potentially corrupted descriptions and selecting the one found to be most reliable. The goal is for the combined $(G, R)$ system to have lower error rate than $G$ alone, and for the error rate to decay quickly with $N$. Our main contributions are as follows:

- We show that when the channel distributions are independent, this simple protocol is asymptotically error-free (*i.e.*, it generates an acceptable answer almost surely when $N \to \infty$), even in scenarios where the reranker is imperfect, *e.g.*, governed by a Mallows or a Zipf-Mandelbrot model. In the former case, the error probability decays exponentially fast (§3).

- We show that the protocol is still asymptotically error-free if we assume that the channel distributions are statistically dependent. When they are coupled by a Beta prior, we show that the error probability decays as a power law when the reranker is perfect (§4).

- We use our framework to obtain "reranking laws", which we validate empirically on text-to-code generation with DeepSeek-Coder 7B (§5.1), on machine translation of medical data with TowerInstruct 13B (§5.2), and on mathematical and commonsense reasoning benchmarks (App. B.3).

**Notation.** We denote $[N] := \{1, ..., N\}$ and we use the shorthand notation $X_{1:N} := (X_1, ..., X_N)$. We use capital letters $(X, Y, ...)$ for random variables and represent probability distributions by $\mathbb{P}(X), \mathbb{P}(Y)$, etc. We denote expectations of functions $f$ under $\mathbb{P}(X)$ by $\mathbb{E}_X[f(X)]$.

## 2 A Communication-Theoretic Perspective of Generator-Reranker Systems

The focus of our paper is on **generator-reranker systems**: a **generator** $G$ (such as an LLM) is prompted with a **query** $q$ (*e.g.*, a question to be answered, a source text to be translated, or a textual prompt for code). As a response to this query, $G$ generates $N$ candidate answers $y_1, ..., y_N$ (called **hypotheses**). We are agnostic about the internals of $G$ and the way the hypotheses are generated: they could come from the same system through sampling or beam search, or they could come from an ensemble of different systems. These hypotheses are then processed by a **reranker** $R$, which ranks them and returns as the final output the one which is found to be the best answer. We are also agnostic about how $R$ is built—it could be an external system or it could be part of (or share parameters with) the generator. Commonly used rerankers are quality estimators (Fernandes et al., 2022), energy-based models (Bhattacharyya et al., 2021), reward models (Li et al., 2022), and minimum Bayes risk decoders (Kumar and Byrne, 2002; Eikema and Aziz, 2020; Freitag et al., 2022; Shi et al., 2022).

Regardless of specific design decisions, the goal of the generator-reranking system $(G, R)$ is to leverage the reranker $R$ to produce better answers (according to some quality metric) than the ones which would be obtained through $G$ alone (*e.g.*, a single sample). In this paper, we show that the propensity for this combined system to generate unacceptable outputs, such as those containing critical errors or hallucinations, decays quickly enough with $N$ under mild assumptions on $G$ and $R$.

We draw an analogy with communication theory as follows. Let $\Sigma$ be an underlying alphabet and $\Sigma^* := \bigcup_{i=0}^{\infty} \Sigma^i$ its Kleene closure, *i.e.*, the set of strings from $\Sigma$. Given the query $q$, we denote by $\mathcal{X}(q) \subseteq \Sigma^*$ the set of **acceptable answers**.[1] We assume the communication system depicted in Fig. 1 (left), a form of **multiple description source coding** (Ozarow, 1980; Gamal and Cover, 1982; Laneman et al., 2005). In this framework, the sender transmits $N$ acceptable answers (called **descriptions**) $x_1, ..., x_N \in \mathcal{X}(q)^N$ in parallel through noisy channels. These descriptions are corrupted according to a distribution $\mathbb{P}(y_1, ..., y_N | x_1, ..., x_N)$, so that some hypotheses $y_i$ may become unacceptable ($y_i \in \Sigma^* \setminus \mathcal{X}(q)$). This "channel noise" is a way to conceptualize the imperfections of the generator $G$. On the receiver side, a decoder processes the (potentially) corrupted descriptions and estimates $\hat{x} = g(y_1, ..., y_N)$ using some decoding function $g$. The overarching goal is to achieve a low error probability $P_{\text{err}}(N; q) := \mathbb{P}(\hat{X} \notin \mathcal{X}(q) \mid q)$ for any query $q$. By bounding the maximal probability of error (over all queries), the average error probability is automatically bounded (Cover and Thomas, 2006, §8). In this paper, we focus on rerankers as the decoding functions, where $g(y_1, ..., y_N)$ returns the top ranked answer, *i.e.*, $g(y_1, ..., y_N) = y_i$ for some $i \in [N]$.

We formalize this construction by considering different models for $G$ and $R$ in the following sections, studying the conditions under which the resulting protocol is **asymptotically error-free**:

> **Definition 1.** *A protocol is asymptotically error-free if, for any query $q$, the probability of the decoder outputting an unacceptable answer approaches zero as $N$ tends to infinity, i.e.,*
>
> $$\lim_{N \to \infty} \underbrace{\mathbb{P}(g(Y_1, ..., Y_N) \notin \mathcal{X}(q) \mid q)}_{:= P_{\text{err}}(N; q)} = 0. \tag{1}$$

For simplicity, we assume that $X_1, ..., X_N$ are conditionally independent given the query $q$, *i.e.*, that $\mathbb{P}(x_1, ..., x_N | q) = \prod_{i=1}^{N} \mathbb{P}(x_i | q)$.[2] We also assume that $Y_{1:N}$ are independent from $q$ given $X_{1:N}$ such that $q \to X_{1:N} \to Y_{1:N}$ forms a Markov chain. Taken together, these two assumptions mean that $\mathbb{P}(x_{1:N}, y_{1:N} | q) = \mathbb{P}(x_{1:N} | q)\mathbb{P}(y_{1:N} | x_{1:N}) = \left( \prod_{i=1}^{N} \mathbb{P}(x_i | q) \right) \mathbb{P}(y_{1:N} | x_{1:N})$.

## 3  Generator-Reranker Systems with Independent Hypotheses

We first consider the case where the corrupted descriptions $Y_{1:N}$ are conditionally independent and identically distributed (i.i.d.) given $X_{1:N}$ and where $Y_i$ depends only on $X_i$, that is, $\mathbb{P}(y_{1:N} | x_{1:N}) = \prod_{i=1}^{N} \mathbb{P}(y_i | x_i)$. Conceptually, this is the scenario where the parallel channels do not interfere, and it corresponds to the graphical model shown in Fig. 1 (right) without the part in red. While this case may not be very realistic in practice—for example, when the hypotheses produced by the generator are all sampled from the same model—it makes the analysis simpler. We will show later in §4 how the analysis can be extended when this assumption does not hold, reusing the results from this section.

In the sequel, given a query $q$, we let $\epsilon$ denote the probability of a hypothesis being unacceptable, $\epsilon := \mathbb{P}(Y_i \notin \mathcal{X}(q) \mid X_i = x_i, q) = \mathbb{P}(Y_i \notin \mathcal{X}(q) \mid X_i = x_i)$.

### 3.1  Perfect and random rerankers

We start by assuming that $R$ is a **perfect reranker**, which implies that it produces an acceptable output when presented with a set of $N$ hypotheses if and only if at least one of them is acceptable. In

---

[1]A key difference between our framework and most lossless communication systems is that there is no need to communicate a *specific* message—any answer in the equivalence class $\mathcal{X}(q)$ is acceptable, hence, if the decoder recovers any message in this set, the communication is considered successful. This is a natural conceptualization in problems involving natural language (where a paraphrase of a correct answer is still correct) or code (where multiple programs might lead to the same execution).

[2]In fact, all results in this paper still hold if there are dependencies between $X_1, ..., X_N$.

this case, the error probability becomes

$$P_{\mathrm{err}}(N; q) = \mathbb{P}(g(Y_1, ..., Y_N) \notin \mathcal{X}(q) \mid q) = \mathbb{E}_{X_{1:N}|q}\big[\mathbb{P}(g(Y_1, ..., Y_N) \notin \mathcal{X}(q) \mid X_{1:N}, q)\big]$$

$$= \mathbb{E}_{X_{1:N}|q}\big[\mathbb{P}(Y_i \notin \mathcal{X}(q), \ \forall i \in [N] \mid X_{1:N})\big]$$

$$= \mathbb{E}_{X_{1:N}|q}\bigg[\prod_{i=1}^{N} \underbrace{P(Y_i \notin \mathcal{X}(q) \mid X_i)}_{=\epsilon}\bigg] = \epsilon^N. \qquad (2)$$

Thus, $P_{\mathrm{err}}(N; q)$ goes to zero exponentially fast with $N$ for any $\epsilon \in [0, 1)$, indicating that when the hypotheses are independent and the reranker is perfect, the protocol is error-free.

On the other end of the spectrum, if the reranker is **random**—*i.e.*, if it selects one of the $N$ hypotheses uniformly at random, we obtain

$$P_{\mathrm{err}}(N; q) = \mathbb{P}(g(Y_1, ..., Y_N) \notin \mathcal{X}(q) \mid q) = \mathbb{E}_{X_{1:N}|q}\big[\mathbb{P}(g(Y_1, ..., Y_N) \notin \mathcal{X}(q) \mid X_{1:N}, q)\big]$$

$$= \mathbb{E}_{X_{1:N}|q}\Big[\mathbb{E}_i\big[\mathbb{P}(Y_i \notin \mathcal{X}(q) \mid X_{1:N}, i)\big]\Big] = \epsilon, \quad (3)$$

that is, we obtain the same error probability as the generator alone, as expected.

### 3.2 Imperfect reranker: Mallows model

We consider now more realistic rerankers. A statistical ranking model widely used in machine learning applications is the **Mallows model** (Klementiev et al., 2008, 2009; Chierichetti et al., 2018; Tang, 2019). Let $\Pi$ denote the set of permutations over $N$ elements, and let $d : \Pi \times \Pi \to \mathbb{R}_+$ be a distance function between permutations. In this paper, we use the Kendall-tau distance $d(\pi, \pi')$, which returns the number of adjacent transpositions needed to turn $\pi$ into $\pi'$. Given a location parameter $\pi_0 \in \Pi$ and a scale parameter $\lambda \in \mathbb{R}_+$, the probability of a ranking $\pi$ according to the Mallows model is $\mathbb{P}(\pi; \pi_0, \lambda) = \exp(-\lambda d(\pi, \pi_0))/Z(\lambda)$, where $Z(\lambda)$ is the partition function.

In our setting, we assume that $\pi_0$ is the ground truth (oracle) ranking[3] of the hypotheses $y_1, ..., y_N$ and $\pi$ is the ranking obtained by the reranker model, so that $\mathbb{P}(\pi; \pi_0, \lambda)$ expresses how imperfect the reranker might be. Note that the family of Mallows models include both perfect and random rerankers as limit cases, respectively as $\lambda \to +\infty$ and as $\lambda = 0$.[4]

Let $\eta_j$ denote the marginal probability that the reranker places at the top the $j^{\mathrm{th}}$ highest ranked hypothesis according to the oracle, *i.e.*, $\eta_j = \mathbb{P}(\pi_0(\pi^{-1}(1)) = j)$. When $K$ out of the $N$ hypotheses are unacceptable, the reranker will pick an unacceptable hypothesis with probability $\sum_{j=N-K+1}^{N} \eta_j$. Combining this with the fact that the probability of $G$ generating $K$ unacceptable hypotheses is a binomial distribution, the error probability becomes

$$P_{\mathrm{err}}(N; q) = \mathbb{P}(g(Y_1, ..., Y_N) \notin \mathcal{X}(q) \mid q) = \mathbb{E}_{X_{1:N}|q}\big[\mathbb{P}(g(Y_1, ..., Y_N) \notin \mathcal{X}(q) \mid X_{1:N}, q)\big]$$

$$= \sum_{K=0}^{N} \bigg[\binom{N}{K}\epsilon^K (1-\epsilon)^{N-K} \sum_{j=N-K+1}^{N} \eta_j\bigg]. \quad (4)$$

Note that (4) holds for **any reranker** with top-1 (marginal) probability mass function $\boldsymbol{\eta} = [\eta_1, ..., \eta_N]$, not only Mallows models. Naively determining $\boldsymbol{\eta}$ would require marginalizing $\mathbb{P}(\pi; \pi_0, \lambda)$ by summing over all permutations $\pi$ satisfying $\pi_0(\pi^{-1}(1)) = j$, which is intractable due to the factorial number of terms involved. Fortunately, tractable combinatorial expressions exist for Mallows models (Fligner and Verducci, 1986; Lebanon and Mao, 2008): the partition function has the compact expression $Z(\lambda) = \prod_{j=1}^{N}(1 - e^{-\lambda j})/(1 - e^{-\lambda})$, and we have (Lebanon and Mao, 2008, Prop. 5):

$$\eta_j = Z^{-1}(\lambda) \sum_{\pi:j=\pi_0(\pi^{-1}(1))} e^{-\lambda d(\pi, \pi_0)} = \frac{e^{-\lambda(j-1)}}{\sum_{r=1}^{N} e^{-\lambda(r-1)}}. \quad (5)$$

---

[3]More specifically, we assume that the hypotheses are ranked according to some quality metric compatible with $\mathcal{X}(q)$, that is, unacceptable answers should be ranked after acceptable answers.

[4]Notably, $e^{-\lambda}$ strictly between 0 and 1 correspond to imperfect rerankers that are better than random. Lower values indicate higher-quality rerankers, making $e^{-\lambda}$ an inverse measure of reranker quality.

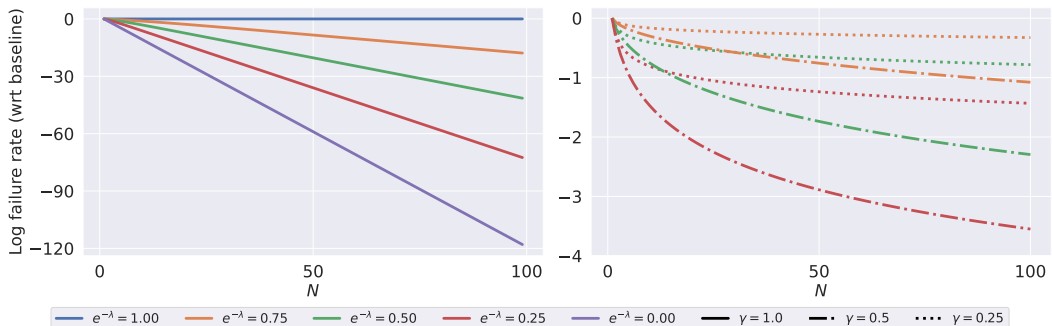

Figure 2: Log of the failure rate (difference with respect to the baseline rate $\log \epsilon$) as a function of the number of generated independent hypotheses $N$ for several values of $e^{-\lambda}$ and $\epsilon = 0.3$. **Left:** Mallows model (§3.2). **Right:** Zipf-Mandelbrot model (§3.3).

Plugging (5) into (4), invoking the binomial theorem, and simplifying, we obtain

$$P_{\mathrm{err}}(N; q) = \mathbb{P}(g(Y_1, ..., Y_N) \notin \mathcal{X}(q) \mid q) = \begin{cases} \epsilon & \text{if } \lambda = 0 \\ \frac{[e^{-\lambda}(1-\epsilon)+\epsilon]^N - e^{-\lambda N}}{1-e^{-\lambda N}} & \text{otherwise.} \end{cases} \tag{6}$$

Notably, when $\lambda \to +\infty$ (perfect reranking), the failure probability becomes $\epsilon^N$, as expected (see (2)), demonstrating the model's ability to interpolate between scenarios of random reranking ($\lambda = 0$) with a failure probability of $\epsilon$ (see (3)), and optimal reranking ($\lambda \to +\infty$) with a failure probability of $\epsilon^N$. A plot is shown in Fig. 2 (left), for several values of $e^{-\lambda} \in [0, 1]$.

Our next result, proved in App. A.1, shows that, even with an imperfect reranker, an asymptotically error-free protocol is possible:

**Proposition 1.** *When $R$ is a Mallows reranker, for any $\lambda > 0$, the protocol is asymptotically error-free and the error probability decays exponentially fast, $P_{\mathrm{err}}(N; q) = \mathcal{O}((e^{-\lambda}(1-\epsilon) + \epsilon)^N)$.*

This result shows that $P_{\mathrm{err}}(N; q)$ converges Q-linearly to zero with rate of convergence $e^{-\lambda}(1-\epsilon) + \epsilon > \epsilon$. Therefore, **Mallows rerankers behave asymptotically as a perfect reranker but where the generator has an increased error probability.**

Given this result, one might wonder whether any reranker "slightly better than random" suffices to obtain an asymptotically error-free protocol. This it **not** the case, as the next counter-example shows.

**Example 1.** *Assume a reranker with probability mass function $\eta_j \propto (N - j + 1)$. The resulting protocol is not asymptotically error-free; we have $P_{\mathrm{err}}(N; q) = \mathcal{O}(\epsilon^2)$. Therefore, the error is reduced from $\mathcal{O}(\epsilon)$ to $\mathcal{O}(\epsilon^2)$ but it is not eliminated. More generally, if $\eta_j \propto (N - j + 1)^r$ for a fixed positive integer $r$, we have $P_{\mathrm{err}}(N; q) = \mathcal{O}(\epsilon^{r+1})$. See App. A.2 for a proof and plots.*

Next, we present a class of rerankers weaker than Mallows which still lead to error-free protocols.

### 3.3 Imperfect reranker: Zipf-Mandelbrot model

For Mallows models (using the Kendall-tau distance), the marginal probabilities (5) can be written as $\boldsymbol{\eta} = \mathrm{softmax}(-\lambda \boldsymbol{z})$, where $\boldsymbol{z} = [0, 1, ..., N - 1]^\top$. We now consider transformations that yield distributions with heavier tails, which we will see later in §5 to be a better empirical fit in several applications. A known extension to softmax is the **$\gamma$-entmax** (Peters et al., 2019),[5] a family of transformations parametrized by $\gamma \geq 0$,

$$\gamma\text{-entmax}(\boldsymbol{z}) := [1 + (\gamma - 1)(\boldsymbol{z} - \tau \boldsymbol{1})]_+^{1/(\gamma-1)}, \tag{7}$$

which recovers softmax as a limit case when $\gamma \to 1$. In (7), $\tau$ is a constant which ensures that $\gamma$-entmax$(\boldsymbol{z})$ is normalized. When $\gamma > 1$, $\gamma$-entmax can return sparse distributions (Blondel et al., 2020). Conversely, when $\gamma < 1$, $\gamma$-entmax leads to heavy-tailed distributions (see App. A.3).

---

[5]Peters et al. (2019) call this $\alpha$-entmax; we use $\gamma$ instead not to clash the notation to be introduced in §4.

Let us now consider $\boldsymbol{\eta} = \gamma\text{-entmax}(-\lambda\boldsymbol{z})$, where $\boldsymbol{z} = [0, 1, ..., N-1]^\top$, instead of (5). Letting $p := 1/(1-\gamma)$, $b = \lambda/p$, and $a = \frac{p+\tau}{\lambda} - 1$ (where $a$ is seen here as a normalizing constant that replaces $\tau$), and assuming $a > -1$ and $\gamma < 1$, we can write the $\gamma$-entmax model as $\eta_j = b^{-p}(a+j)^{-p}$. Note that $\gamma < 1$ is equivalent to $p > 1$. This is called a **Zipf-Mandelbrot model** (Zipf, 1932; Mandelbrot, 1965). This model generalizes the famous Zipf's law, which applies empirically to many practical contexts, such as the frequency table of words in a corpus of natural language (Powers, 1998). The constant $a$ is determined to satisfy $\sum_{j=1}^{N} (a+j)^{-p} = b^p$. When $N \to \infty$, the left hand side becomes the Hurwitz zeta function (Hurwitz, 1882), which equals the Riemann's zeta when $a = 0$,

$$\zeta(p, a+1) := \sum_{j=1}^{\infty} \frac{1}{(a+j)^p} = \frac{1}{\Gamma(p)} \int_0^\infty dt \frac{t^{p-1}}{e^{(a+1)t}(1-e^{-t})}. \tag{8}$$

The following result, proved in App. A.4, shows that Zipf-Mandelbrot rerankers (which are weaker than Mallows rerankers and become the latter when $\gamma \to 1$) still ensure error-free protocols. The proof makes use of the integral representation of the Hurwitz zeta function (8) and of the dominated convergence theorem, reusing the result for Mallows models in Proposition 1.

**Proposition 2.** *When $R$ is a Zipf-Mandelbrot reranker, for any $\lambda > 0$ and $\gamma < 1$, the protocol is asymptotically error-free.*

Fig. 2 (right) shows how this model differs from the one presented in §3.2. Since the reranker is weaker, the error curves bend causing the error decrease to be slower, but still convergent to zero.

## 4 Generator-Reranker Systems with Dependent Hypotheses

We assume now a more realistic scenario where the independence assumption of §3 might not hold. For example, $(X_1, Y_1), ..., (X_N, Y_N)$ might be only **exchangeable**—this is the case, for example, when the hypotheses are generated from $G$ by sampling from a given model, conditioned on the query. In communication theory parlance, this assumes the presence of channel "interference" that introduces dependencies between the errors at the various channels, although permuting the messages at each channel does not change the joint distribution. By de Finetti's theorem (Diaconis and Freedman, 1980), exchangeability implies that there is some mixture variable $h \in \mathcal{H}$ such that $\mathbb{P}(x_{1:N}, y_{1:N}) = \int_{\mathcal{H}} d\mathbb{P}(h) \prod_{i=1}^{N} \mathbb{P}(x_i|h)\mathbb{P}(y_i|x_i, h)$.

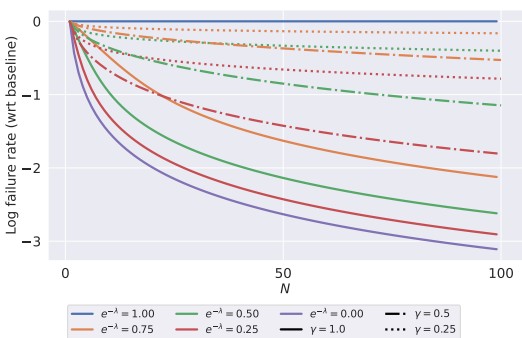

Figure 3: Log of the failure rate as a function of the number of generated **exchangeable** hypotheses $N$ for several values of $\gamma$, $e^{-\lambda}$, and $\epsilon = \alpha = 0.3$.

We assume further that $h = (q, \tau)$ can be decoupled into the query variable $q$, which conditions $x$, and a random variable $\tau$, which conditions $y$, such that $\mathbb{P}(x_i|h) := \mathbb{P}(x_i|q)$ and $\mathbb{P}(y_i|x_i, h) := \mathbb{P}(y_i|x_i, \tau)$. This corresponds to the graphical model in Fig. 1 (right), including the part in red. We let $\tau$ be a continuous random variable in $[0, 1]$ such that $\mathbb{E}[\tau] = \epsilon = \mathbb{P}(Y_i \neq \mathcal{X}(q) \mid X_i)$. A convenient choice is a Beta distribution with parameters $\alpha$ and $\beta$, $p(\tau; \alpha, \beta) := \frac{\Gamma(\alpha+\beta)}{\Gamma(\alpha)\Gamma(\beta)} \tau^{\alpha-1}(1-\tau)^{\beta-1}$.

**Perfect reranker and Beta coupling.** If $R$ is a perfect reranker, the error probability is

$$P_{\text{err}}(N; q) = \mathbb{P}(g(Y_1, ..., Y_N) \notin \mathcal{X}(q) \mid q) = \mathbb{E}_{X_{1:N}|q}\big[\mathbb{P}(g(Y_1, ..., Y_N) \notin \mathcal{X}(q) \mid X_{1:N}\big]$$

$$= \mathbb{E}_{X_{1:N}}\left[\int_0^1 d\tau\, p(\tau) \prod_{i=1}^{N} \underbrace{\mathbb{P}(Y_i \notin \mathcal{X}(q) \mid X_i, \tau)}_{=\tau}\right] = \mathbb{E}_\tau[\tau^N]. \tag{9}$$

When $\tau \sim \text{Beta}(\tau; \alpha, \beta)$, the $N^\text{th}$-raw moment (9) has a closed form, leading to $P_{\text{err}}(N; q) = \prod_{i=1}^{N} \frac{\alpha+i-1}{\alpha+\beta+i-1}$. The next result, proved in App. A.5 using Gautschi's inequality (Gautschi, 1959) and the Stirling's formula, shows that we still obtain an error-free protocol, albeit the error decays slower than in the independent case—no longer exponentially but rather following a power law.

**Proposition 3.** *When $\tau \sim Beta(\tau; \alpha, \beta)$ and with a perfect reranker, the protocol is error-free and the error probability decays as a power law, $P_{\mathrm{err}}(N; q) = \mathcal{O}(N^{-\beta})$. Furthermore, for $\beta < 1$, we have $P_{\mathrm{err}}(N; q) \in \left( \frac{\Gamma(\alpha+\beta)}{\Gamma(\alpha)}(\alpha + \beta + N)^{-\beta}, \frac{\Gamma(\alpha+\beta)}{\Gamma(\alpha)}(\alpha + \beta + N - 1)^{-\beta} \right)$.*

**Imperfect reranker.** When $\tau \sim \mathrm{Beta}(\tau; \alpha, \beta)$, the probability of exactly $K$ out of $N$ messages being corrupted is (due to the conjugacy between the Beta prior and the binomial distribution) $\binom{N}{K} \int_0^1 d\tau \; p(\tau; \alpha, \beta) \tau^K (1 - \tau)^{N-K} = \binom{N}{K} \frac{\prod_{i=1}^K (\alpha+i-1) \prod_{i=1}^{N-K} (\beta+i-1)}{\prod_{i=1}^N (\alpha+\beta+i-1)}$. Therefore, using the reranker marginals $\boldsymbol{\eta}$ as in (4), we get

$$P_{\mathrm{err}}(N; q) = \sum_{K=0}^N \binom{N}{K} \frac{\prod_{i=1}^K (\alpha + i - 1) \prod_{i=1}^{N-K}(\beta + i - 1)}{\prod_{i=1}^N (\alpha + \beta + i - 1)} \sum_{j=N-K+1}^N \eta_j, \qquad (10)$$

which leads to the plot in Fig. 3 for Mallows and Zipf-Mandelbrot models.[6]

The next result, proved in App. A.6, shows that the dependencies considered in this subsection do not compromise the error-free protocol when it exists for any density $p(\tau)$ which is finite in $(0, 1)$ (not necessarily a Beta distribution). The proof invokes the dominated convergence theorem to enable commuting the limit with the integral sign.

**Proposition 4.** *Let $G_\tau$ be a generator producing independent hypotheses (§3) where each hypothesis is acceptable with probability $1 - \tau$. Let the reranker $R$ be such that $(G_\tau, R)$ has error probability $P_{\mathrm{err}}^{\mathrm{indep}}(N; q, \tau) \to 0$ for every $\tau \in (0, 1)$ (i.e., it is asymptotically error-free). Assume that the function $\tau \mapsto P_{\mathrm{err}}^{\mathrm{indep}}(N; q, \tau)$ is measurable for every $N \in \mathbb{N}$. Then, when $R$ is used with a generator $G$ which produces exchangeable hypotheses with arbitrary distribution $p(\tau)$, finite in $(0, 1)$, the system $(G, R)$ is still asymptotically error-free.*

This result has important implications: it tells us that, to design error-free protocols, it is sufficient to verify if they are error-free in the simpler case where hypotheses are independent.

## 5 Experiments

In this section, we demonstrate the validity of our reranking laws on two different tasks:[7] text-to-code generation (§5.1) and machine translation of medical data (§5.2). Following existing literature on scaling laws for language modeling, we fit all curves on the development set using least squares (Ghorbani et al., 2022, App. E) and plot them on the *unseen* test set.[8] In all cases, we consider the generalized model presented in §4 with parameters $\alpha$, $\beta$, and a Zipf-Mandelbrot reranker with parameters $\gamma$, and $e^{-\lambda}$, which becomes a Mallows reranker when $\gamma \to 1$. This is done in two steps: first, we fit $\alpha$ and $\beta$ using the data for the perfect reranker ($e^{-\lambda} = 0$). Then, we fit $\gamma$ and $e^{-\lambda}$ using the already estimated $\alpha$ and $\beta$ and the data for the imperfect reranker. Our code is available at https://github.com/deep-spin/reranking-laws.

### 5.1 Code generation

We use a sanitized version of the MBPP dataset (Austin et al., 2021; Liu et al., 2023), a widely used benchmark for evaluating code LLMs, which includes 400 programming problems in Python. For each problem, the dataset includes ground-truth programs and three test cases with input and ground-truth output. We split the dataset in two equally sized parts to get development and test splits.

We generate 200 hypotheses with DeepSeek-Coder 7B (Guo et al., 2024) using a sampling temperature of 1 (see App. B.1 for the prompt template). As in previous work, for simplicity, we use only one test case for each problem (Shi et al., 2022), and select one candidate by taking a **majority vote** over the

---

[6]Since $\tau \sim \mathrm{Beta}(\tau; \alpha, \beta)$, we have $\mathbb{E}[\tau] = \alpha/(\alpha + \beta)$, which we set to $\epsilon$ to match the independent setting from §3, resulting in $\beta = (\epsilon^{-1} - 1)\alpha$. Hence, $\alpha$ is our only new free parameter. As $\alpha \to 0^+$, the hypotheses become maximally dependent and reranking is hopeless; as $\alpha \to \infty$, the scenario reverts to full independence.

[7]App. B.3 contains additional experiments on mathematical and commonsense reasoning benchmarks.

[8]We use `scipy.optimize.least_squares`.

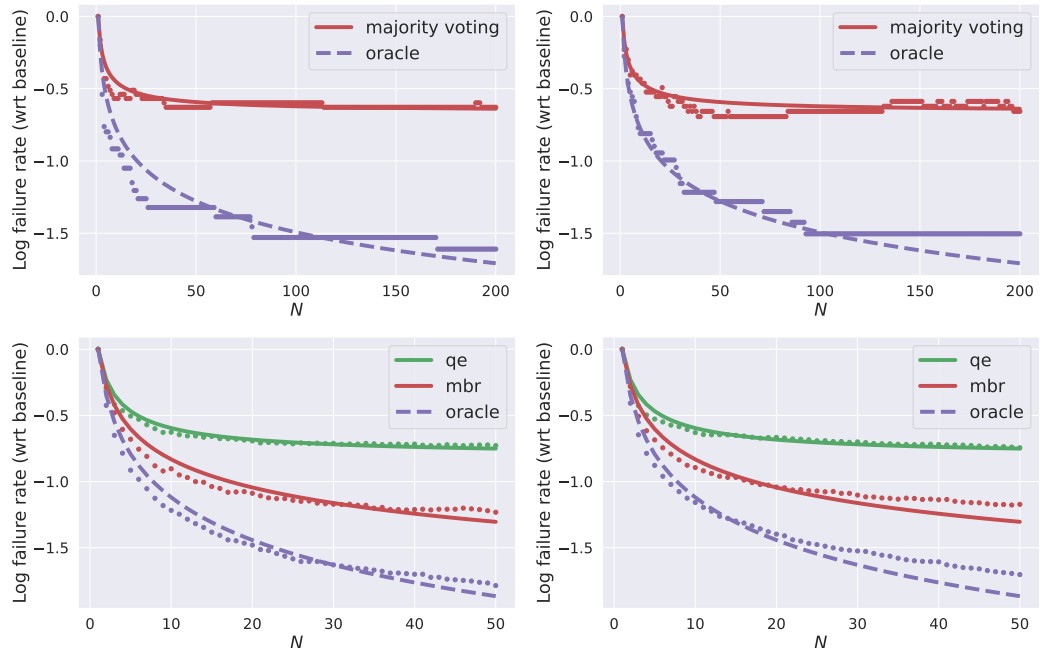

Figure 4: Log of the failure rate as a function of $N$. The empirical data is represented with dots (**left:** dev, **right:** test set) and our fitted models with solid and dashed lines (imperfect and perfect reranker, respectively). **Top:** text-to-code generation (§5.1). **Bottom:** machine translation (§5.2).

execution results, dismissing hypotheses that fail to execute on the test case (Wang et al., 2023). A hypothesis is considered unacceptable if the result of at least one test case (out of three) is different from the ground truth.

Fig. 4 (top) shows the log failure rate on the dev and test sets (left and right, respectively) as a function of $N$. Even though the oracle fit is not perfect, we get $\alpha = .1$, $\beta = .309$, $\gamma = .001$, and $e^{-\lambda} = .003$ for the imperfect reranker with majority voting, which fits the data well, as shown by the red curve.

## 5.2 Machine translation

We use the TICO-19 dataset (Anastasopoulos et al., 2020), which includes 3071 English sentences in the medical domain (*i.e.*, COVID-19 related content) translated into 38 languages. We use the official splits, which contain 971 examples for development and 2100 for testing, focusing on translating from English (EN) to Portuguese (PT), Spanish (ES), and Russian (RU).

For each source sentence, we sample 50 translation hypotheses with a temperature of 1 from TowerInstruct 13B (Alves et al., 2024) using the prompt template in App. B.2.[9] Following Farinhas et al. (2023), we consider two reranking strategies: selecting the best candidate with **MBR decoding** using COMET-22 as the utility metric (Eikema and Aziz, 2020; Rei et al., 2022a) and **reranking based on quality estimation** using the reference-free CometKiwi (Fernandes et al., 2022; Rei et al., 2022b). Since we cannot afford to collect human evaluation scores for each sampled hypothesis, we consider a translation to have a critical mistake (*i.e.*, to be unacceptable) if its COMET-22 score is below 0.85, and an **oracle** (perfect) reranker that picks the translation with the highest COMET-22 score.

We follow the described procedure using the data from all language pairs together. Fig. 4 (bottom) shows the log failure rate on the dev and test sets as a function of $N$. We get $\alpha = 0.1$ and $\beta = 0.46$. Additionally, we have $\gamma = 0.182$ and $e^{-\lambda} = 0.001$ for MBR decoding and $\gamma = 0.001$ and $e^{-\lambda} = 0.005$ for QE reranking. See App. B.2 for additional plots showing these curves when the data

---

[9]This model outperforms all existing open-source alternatives (even of larger scales) for translating content between the supported languages and is also competitive with GPT-4 (OpenAI et al., 2023), especially when combined with MBR decoding (Alves et al., 2024, App. A).

from each language pair is used to fit a separate model. Again, we see a reasonable fit, especially for the imperfect rerankers, with MBR decoding leading to lower failure rates than reranking with QE.

# 6  Discussion and Related Work

We believe the communication-theoretic perspective introduced in this paper might inspire the design of new protocols for increasing the quality and safety of LLMs. The generator-reranker system studied in this paper bears resemblance with repetition codes, a very naive (and inefficient) class of error-correcting codes. Can more powerful designs (Hamming, 1950; Reed and Solomon, 1960; Gallager, 1962; Berrou et al., 1993) inspire more efficient protocols? In machine translation, other forms of adding redundancy, such as lattice generation (Singhal et al., 2023) and hypothesis recombination (Vernikos and Popescu-Belis, 2024), suggest that more efficient designs are indeed possible.

Recent work also suggests that **LLM-based evaluators** could be used as highly effective rerankers in specific tasks (Kim et al., 2024). While LLMs are not yet ready to fully replace human evaluators across diverse NLP tasks (Bavaresco et al., 2024), in some cases, they can even provide fine-grained assessments in addition to single scores (Kocmi and Federmann, 2023; Fernandes et al., 2023a).

Another class of communication systems allow for **feedback**, *e.g.*, in "automatic repeat request" protocols (Lin et al., 1984), where the receiver has a backchannel to request the sender to retransmit missing bits of information. This framework can be useful to analyze LLM protocols where the generator generates a varying number of hypotheses interactively, relying on feedback from another module, such as a reward model or a confidence estimator, as in Quach et al. (2023). Communication with feedback was also used recently by Jung et al. (2024) for summarization when the generator error probability $\epsilon$ is large—our mild conditions for asymptotically error-free protocols (Propositions 1–4) suggest that "bootstrapping" a correct answer is possible even in scenarios where $G$ is very weak. Additionally, recent work has shown that LLMs may struggle with planning or self-verification, advocating instead for tighter integration between LLMs and external model-based verifiers (Kambhampati et al., 2024). This supports our view that using external feedback models can improve LLMs by enabling interactive, error-correcting communication.

We provide **reranking laws**, which allow us to predict how many hypotheses are necessary to achieve a desired error probability. This links to a rich body of literature aiming to predict the performance of deep learning models in terms of fundamental parameters, such as the model size or the amount of compute and data used to train them (Hestness et al., 2017, 2019). These so called "neural scaling laws" have been studied in the context of language modeling (Kaplan et al., 2020; Hoffmann et al., 2022) and machine translation (Ghorbani et al., 2022; Fernandes et al., 2023b), where we observe a power-law scaling for the performance as a function of each fundamental parameter. Our paper complements this line of work by considering the decoding dimension for generator-reranker systems.

The analysis and theoretical results of this paper focus on binary acceptable/unacceptable decisions; however it is possible to extend our framework to consider also **continuous quality metrics** (such as COMET scores for translation (Rei et al., 2020)) by replacing the notion of "asymptotically error-free" protocol (Definition 1) by a more general concept associated to a quality target. A possible path is to posit a probability *density* for the continuous quality metric (instead of a Bernoulli error probability) for each hypothesis coming from the generator, such as a Gaussian or uniform distribution with some input-dependent parameters. For a perfect reranker and independent hypotheses, the resulting output after reranking would be distributed according to the corresponding *extreme value distribution* (this models the distribution of the *highest* evaluation metric score among the $N$ hypotheses). Extreme value distributions are an important subject of study in order statistics (David and Nagaraja, 2004) and their densities have closed form expressions in some restricted cases: for example, the Gaussian assumption above yields a Gumbel distribution, and a uniform assumption yields a Beta distribution. The asymptotic case ($N \to \infty$) corresponds to one of Gumbel, Fréchet or Weibull families (this is a consequence of the Fisher–Tippett–Gnedenko theorem (David and Nagaraja, 2004)). From the extreme value distribution, we can obtain the *expected* evaluation metric score or the probability of a quality score being below an acceptable threshold. However, the generalization to imperfect rerankers (such as the Mallows or Zipf-Mandelbrot rerankers described in §3.2 and 3.3) seems harder than in the binary case and requires further investigation.

# 7 Conclusions

We presented a communication-theoretic perspective of generator-reranker LLMs, where the generator $G$ is conceptualized as a sender transmitting $N$ descriptions in parallel through noisy channels, and the reranker $R$ decodes the message by selecting the most appropriate description. Under mild conditions, the combined system $(G, R)$ yields an acceptable answer almost surely when $N \to \infty$. Experiments on text-to-code generation and machine translation with LLMs validate our framework.

# 8 Limitations and Broader Impacts

We regard our paper as a first step connecting communication theory and LLMs, as discussed in §6. However, it should be noted that our work has several limitations. First, the guarantees of error-free protocol in Propositions 1–4 are only asymptotic, and in certain cases a large $N$ may be necessary to achieve a large enough error decrease. We provide convergence rates only for Mallows rerankers (with independent hypotheses and also in the dependent case, when combined with a Beta prior). Second, there is no simple recipe to determine if the Mallows and Zipf-Mandelbrot reranker models are a good empirical fit to concrete rerankers. The same applies to the prior distribution $p(\tau)$ which makes hypotheses dependent. Third, while our experiments in §5 suggest a reasonable fit in two tasks (code generation and machine translation), the fit is not perfect. A challenge is that, for large $N$, errors are rare events, and therefore prone to statistical inaccuracies (this is visible in the "steps" observed in the code generation plots). Finally, although our framework focuses on binary acceptable/unacceptable decisions, it can be extended to continuous evaluation metrics, but this would require modifications to some concepts (*e.g.*, the notion of asymptotically error-free protocols). Despite these limitations, the binary case remains highly relevant in practice—for example, in code generation, where the output either executes correctly or it does not. We expect future work to overcome some of these limitations.

In considering the broader impact of our work, it is crucial to acknowledge its early stage and predominantly theoretical nature, which lends the discussion a speculative quality. We believe that our research can significantly enhance the reliability of LLMs by facilitating the identification of potential system failures, holding promise in fields such as natural language processing and computer vision, where robustness and error prediction are paramount. While not directly addressing environmental concerns shared across different LLMs (Strubell et al., 2019), our work could indirectly contribute to energy efficiency efforts by quantifying the efficiency of reranking methods, potentially reducing computational requirements while maintaining requisite quality thresholds during inference.

## Acknowledgments

We would like to thank Ben Peters, Duarte Alves, Marcos Treviso, Mário Figueiredo, Sweta Agrawal, and the SARDINE lab team for helpful discussions. This work was supported by EU's Horizon Europe Research and Innovation Actions (UTTER, contract 101070631), by the project DECOLLAGE (ERC-2022-CoG 101088763), by the Portuguese Recovery and Resilience Plan through project C645008882-00000055 (Center for Responsible AI), and by Fundação para a Ciência e Tecnologia through contract UIDB/50008/2020.

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

## A  Proofs and Visualizations

### A.1  Proof of Proposition 1

Let $\lambda_\epsilon := -\log\left(e^{-\lambda}(1-\epsilon)+\epsilon\right)$ and define $F(N) = \log P_{\mathrm{err}}(N;q) = \frac{e^{-\lambda_\epsilon N}-e^{-\lambda N}}{1-e^{-\lambda N}}$. Observe that $0 < \lambda_\epsilon < \lambda$ for any $\lambda > 0$ and $\epsilon \in (0,1)$. We extend the domain of $F$ to the real numbers in $[1,+\infty)$. We will prove that $F(N)$ is decreasing and that $\lim_{N\to\infty} F'(N) = -\lambda_\epsilon$. This shows that $P_{\mathrm{err}}(N;q) \to 0$ at asymptotic rate $e^{-\lambda_\epsilon}$. We have

$$F'(N) = \frac{(e^{-\lambda_\epsilon N}-e^{-\lambda N})'}{e^{-\lambda_\epsilon N}-e^{-\lambda N}} - \frac{(1-e^{-\lambda N})'}{1-e^{-\lambda N}} = \frac{-\lambda_\epsilon e^{-\lambda_\epsilon N}+\lambda e^{-\lambda N}}{e^{-\lambda_\epsilon N}-e^{-\lambda N}} - \frac{\lambda e^{-\lambda N}}{1-e^{-\lambda N}} \le 0,$$

hence $F(N)$ is decreasing. Since the second term tends to zero, the limit is given by the first term:

$$\lim_{N\to\infty} F'(N) = \lim_{N\to\infty} \frac{-\lambda_\epsilon e^{-\lambda_\epsilon N}+\lambda e^{-\lambda N}}{e^{-\lambda_\epsilon N}-e^{-\lambda N}} = \lim_{N\to\infty} \frac{-\lambda_\epsilon}{1-e^{(-\lambda+\lambda_\epsilon)N}} + \frac{\lambda}{e^{(-\lambda_\epsilon+\lambda)N}-1} = -\lambda_\epsilon,$$

where we used the fact that $e^{(-\lambda+\lambda_\epsilon)N} \to 0$ and $e^{(-\lambda_\epsilon+\lambda)N} \to +\infty$. This proves the desired claim, that is, the error probability decreases exponentially fast with rate $e^{-\lambda_\epsilon}$. Note that, for a perfect reranker ($\lambda \to \infty$), we get $e^{-\lambda_\epsilon} = \epsilon$ and we recover the rate $\epsilon^N$ seen in §3.1.

### A.2  Proof of Example 1

We first provide a proof for $r = 1$. We have $\sum_{j=N-K+1}^{N} \eta_j = \sum_{j=1}^{K} \eta_{N-K+j} = \frac{\sum_{j=1}^{K} j}{\sum_{j=1}^{N} j} = \frac{K(K+1)}{N(N+1)}$. Plugging this into Eq. (4), we obtain

$$P_{\mathrm{err}}(N;q) = \sum_{K=0}^{N} \binom{N}{K} \epsilon^K (1-\epsilon)^{N-K} \frac{K^2+K}{N^2+N} = \frac{\mathbb{E}_{K\sim B(N,\epsilon)}[K^2+K]}{N^2+N}$$

$$= \frac{N\epsilon(1-\epsilon)+N^2\epsilon^2+N\epsilon}{N(N+1)} = \frac{\epsilon(1-\epsilon)+N\epsilon^2+\epsilon}{N+1}, \tag{11}$$

where $B(N,\epsilon)$ denotes the binomial distribution with parameters $N$ and $\epsilon$ and we use the facts that $\mathbb{E}_{K\sim B(N,\epsilon)}[K] = N\epsilon$ and $\mathbb{E}_{K\sim B(N,\epsilon)}[K^2] = N\epsilon(1-\epsilon)+N^2\epsilon^2$. Therefore, we obtain $\lim_{N\to\infty} P_{\mathrm{err}}(N;q) = \epsilon^2$.

We now prove the general case $r \ge 1$. From Faulhaber's formula, we have $\sum_{j=1}^{K} j^r = \frac{1}{r+1}\sum_{j=0}^{r}\binom{r+1}{j}B_j K^{r-j+1}$, where $B_j = \sum_{\ell=0}^{j}\frac{1}{\ell+1}\sum_{m=0}^{\ell}\binom{\ell}{m}(-1)^m(m+1)^j$ denotes the $j^{\mathrm{th}}$ Bernoulli number. Therefore, we get

$$\sum_{j=N-K+1}^{N} \eta_j = \sum_{j=1}^{K} \eta_{N-K+j}^r = \frac{\sum_{j=1}^{K} j^r}{\sum_{j=1}^{N} j^r} = \frac{\sum_{j=0}^{r}\binom{r+1}{j}B_j K^{r-j+1}}{\sum_{j=0}^{r}\binom{r+1}{j}B_j N^{r-j+1}}. \tag{12}$$

Plugging this into Eq. (4), we obtain

$$P_{\mathrm{err}}(N;q) = \sum_{K=0}^{N} \binom{N}{K} \epsilon^K (1-\epsilon)^{N-K} \frac{\sum_{j=0}^{r}\binom{r+1}{j}B_j K^{r-j+1}}{\sum_{j=0}^{r}\binom{r+1}{j}B_j N^{r-j+1}}$$

$$= \frac{\sum_{j=0}^{r}\binom{r+1}{j}B_j \mathbb{E}_{K\sim B(N,\epsilon)}[K^{r-j+1}]}{\sum_{j=0}^{r}\binom{r+1}{j}B_j N^{r-j+1}}. \tag{13}$$

We now use the fact that the raw moments of the binomial distribution $B(N,\epsilon)$ are given by $\mathbb{E}_{K\sim B(N,\epsilon)}[K^m] = \sum_{\ell=0}^{m}\left\{{m \atop \ell}\right\}N^{\underline{\ell}}\epsilon^\ell$, where $\left\{{m \atop \ell}\right\} := \frac{1}{\ell!}\sum_{i=0}^{\ell}(-1)^{\ell-i}\binom{\ell}{i}i^m$ are the Stirling numbers of the second kind, and $N^{\underline{\ell}} := \frac{N!}{(N-\ell)!}$ is the $\ell^{\mathrm{th}}$ falling power of $N$. Therefore, when $N \to \infty$, (13) becomes

$$\lim_{N\to\infty} P_{\mathrm{err}}(N;q) = \lim_{N\to\infty} \frac{\binom{r+1}{0}B_0 \overbrace{\left\{{r+1 \atop r+1}\right\}}^{=1} N^{\underline{r+1}}\epsilon^{r+1}}{\binom{r+1}{0}B_0 N^{r+1}} = \epsilon^{r+1}. \tag{14}$$

The plots in Fig. 5 show examples for several values of $r$.

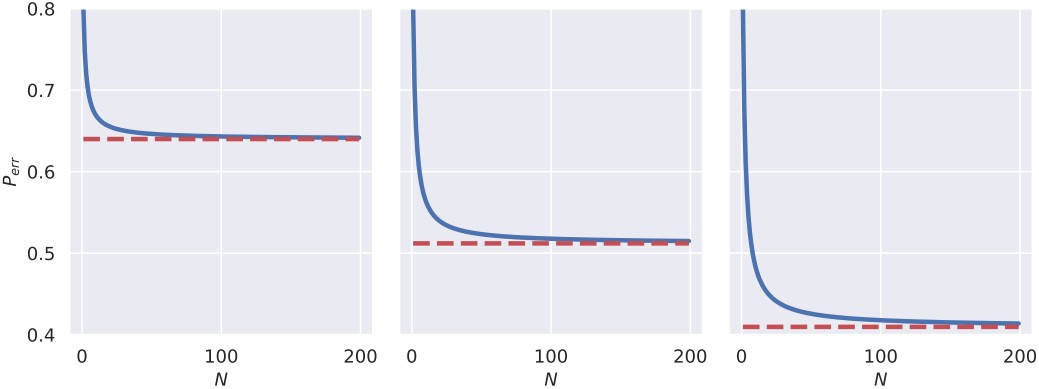

Figure 5: $P_{\text{err}}$ using rerankers with probability mass function $\eta_j \propto (N - j + 1)^r$ with $r = \{1, 2, 3\}$ (from left to right) and $\epsilon = 0.8$. The resulting protocol is not asymptotically error-free: the horizontal asymptotes in red correspond to $\epsilon^{r+1}$, according to Eq. (14).

### A.3 Entmax

When $\gamma > 1$, $\gamma$-entmax can return sparse distributions (Blondel et al., 2020). This case has been extensively studied as a way to, *e.g.*, filter large output spaces (Correia et al., 2020; Peters and Martins, 2021) or to produce more interpretable predictions (Correia et al., 2019; Martins et al., 2020, 2021, 2022). Conversely, when $\gamma < 1$, $\gamma$-entmax leads to distributions with heavier tails, which is the case of our interest, as described in §3.3. See Fig. 6 for an illustration of $\gamma$-entmax for different values of $\gamma$ in the two-dimensional case. For $\gamma > 1$, all mappings saturate at $z = \pm 1/\gamma - 1$; this does not happen for $\gamma \leq 1$.

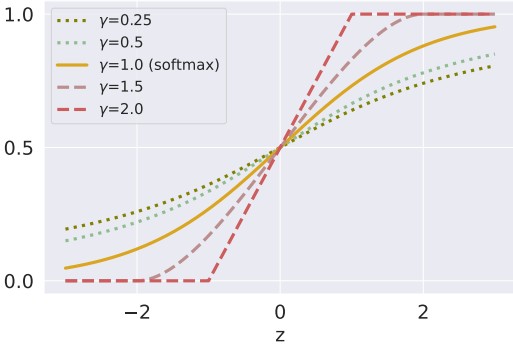

Figure 6: Two-dimensional $\gamma$-entmax$([z, 0])_1$.

### A.4 Proof of Proposition 2

Note that we can write

$$\sum_{j=1}^{N} \frac{1}{(a + j)^p} = \zeta(p, a + 1) - \zeta(p, a + N + 1)$$

and

$$\sum_{j=N-K+1}^{N} \eta_j = b^{-p}(\zeta(p, a + 1) - \zeta(p, a + N + 1) - \zeta(p, a + 1) + \zeta(p, a + N - K + 1))$$

$$= b^{-p}(\zeta(p, a + N - K + 1) - \zeta(p, a + N + 1))$$

$$= \frac{1}{b^p \Gamma(p)} \int_0^\infty dt \frac{t^{p-1}}{e^{(a+N+1)t}(1 - e^{-t})}(e^{Kt} - 1). \tag{15}$$

The error probability is

$$
\begin{aligned}
P_{\text{err}}(N; q) &= \sum_{K=0}^{N} \left[ \binom{N}{K} \epsilon^K (1 - \epsilon)^{N-K} \sum_{j=N-K+1}^{N} \eta_j \right] \\
&= \frac{1}{b^p \Gamma(p)} \int_0^\infty dt \frac{t^{p-1}}{e^{(a+N+1)t}(1 - e^{-t})} \underbrace{\sum_{K=0}^{N} \binom{N}{K} \epsilon^K (1 - \epsilon)^{N-K} (e^{Kt} - 1)}_{=(e^t \epsilon + 1 - \epsilon)^N - 1} \\
&= \frac{1}{b^p \Gamma(p)} \int_0^\infty dt \frac{t^{p-1}[(e^t \epsilon + 1 - \epsilon)^N - 1]}{e^{(a+N+1)t}(1 - e^{-t})} \\
&= \frac{1}{b^p \Gamma(p)} \int_0^\infty dt \frac{t^{p-1}}{e^{(a+1)t}(1 - e^{-t})} \frac{(e^t \epsilon + 1 - \epsilon)^N - 1}{e^{tN}} \\
&= \frac{1}{b^p \Gamma(p)} \int_0^\infty dt \frac{t^{p-1}}{e^{(a+1)t}(1 - e^{-t})} \underbrace{[((1 - \epsilon)e^{-t} + \epsilon)^N - e^{-tN}]}_{:=f_N(t) \to 0}.
\end{aligned} \tag{16}
$$

Since $a$ is the normalizing constant such that $1 = \zeta(p, a+1) = \frac{1}{b^p \Gamma(p)} \int_0^\infty dt \frac{t^{p-1}}{e^{(a+1)t}(1-e^{-t})}$ (cf. Eq. (8)), we can interpret the expression above as the expectation of $f_N(t) := ((1-\epsilon)e^{-t}+\epsilon)^N - e^{-tN}$ under the probability distribution on $(0, \infty)$ with density $\pi(t) := \frac{1}{b^p \Gamma(p)} \frac{t^{p-1}}{e^{(a+1)t}(1-e^{-t})}$. Since $f_N(t) \to 0$ pointwise for $t \in ]0, \infty[$ and it is bounded in that interval, we can invoke the dominated convergence theorem to commute the limit and integral sign. We then have that $P_{\text{err}}(N; q) \to 0$.

### A.5 Proof of Proposition 3

Let us consider first the case where $\beta = 1$. Then,

$$
P_{\text{err}}(N; q) = \prod_{i=1}^{N} \frac{\alpha + i - 1}{\alpha + \beta + i - 1} = \frac{\alpha}{\alpha + 1} \frac{\alpha + 1}{\alpha + 2} \cdots \frac{\alpha + N - 1}{\alpha + N} = \frac{\alpha}{\alpha + N} \to 0.
$$

Now consider the case where $\beta > 1$. We have for each term in the product that $\frac{\alpha+i-1}{\alpha+\beta+i-1} < \frac{\alpha+i-1}{\alpha+i}$, hence we must have $P_{\text{err}}(N; q) < \frac{\alpha}{\alpha+N}$. Since the sequence is positive (since all terms are positive) and decreasing (since all terms are $< 1$), we must also have $P_{\text{err}}(N; q) \to 0$ when $\beta > 1$.

Finally, let us analyze the case where $0 < \beta < 1$. From (9), we have

$$
\begin{aligned}
P_{\text{err}}(N; q) = \mathbb{E}_\tau[\tau^N] &= \int_0^1 \frac{\Gamma(\alpha + \beta)}{\Gamma(\alpha)\Gamma(\beta)} \tau^{\alpha-1}(1 - \tau)^{\beta-1} \tau^N \\
&= \frac{\Gamma(\alpha + \beta)}{\Gamma(\alpha)} \frac{\Gamma(\alpha + N)}{\Gamma(\alpha + \beta + N)} \underbrace{\int_0^1 \frac{\Gamma(\alpha + \beta + N)}{\Gamma(\alpha + N)\Gamma(\beta)} \tau^{\alpha+N-1}(1 - \tau)^{\beta-1}}_{=1} \\
&= \frac{\Gamma(\alpha + \beta)}{\Gamma(\alpha)} \frac{\Gamma(\alpha + N)}{\Gamma(\alpha + \beta + N)}.
\end{aligned} \tag{17}
$$

We invoke Gautschi's inequality, which states that $x^{1-s} < \frac{\Gamma(x+1)}{\Gamma(x+s)} < (x+1)^{1-s}$ for any $x$ and $s \in (0, 1)$. We set $s := 1 - \beta$ and $x := \alpha + \beta + N - 1$, from which we obtain the desired result.

To show that the error probability decays as a power law for any $\beta > 0$, we use Stirling's formula, which states that

$$
\Gamma(z) = \sqrt{\frac{2\pi}{z}} \left(\frac{z}{e}\right)^z \left(1 + \mathcal{O}\left(\frac{1}{z}\right)\right). \tag{18}
$$

Therefore,

$$\lim_{N\to\infty} \frac{\Gamma(\alpha+N)}{\Gamma(\alpha+\beta+N)} = \lim_{N\to\infty} \frac{\sqrt{\frac{2\pi}{\alpha+N}}\left(\frac{\alpha+N}{e}\right)^{\alpha+N}}{\sqrt{\frac{2\pi}{\alpha+\beta+N}}\left(\frac{\alpha+\beta+N}{e}\right)^{\alpha+\beta+N}}$$

$$= \lim_{N\to\infty} \underbrace{\sqrt{\frac{\alpha+\beta+N}{\alpha+N}}}_{\to 1}\ \underbrace{\left(\frac{\alpha+N}{\alpha+\beta+N}\right)^{\alpha+N}}_{\to e^{-\beta}}\left(\frac{\alpha+\beta+N}{e}\right)^{-\beta}$$

$$= \lim_{N\to\infty}(\alpha+\beta+N)^{-\beta} = \mathcal{O}(N^{-\beta}). \tag{19}$$

### A.6 Proof of Proposition 4

Let $P_{\mathrm{err}}^{\mathrm{indep}}(N;q,\tau)$ denote the error probability of the generator-reranker system when the hypotheses are independent and each hypothesis produced by $G$ has error probability $\tau$. The error probability of the generator-reranker system with exchangeable hypotheses is given by

$$P_{\mathrm{err}}(N;q) = \mathbb{P}(g(Y_1,...,Y_N) \notin \mathcal{X}(q) \mid q) = \mathbb{E}_{X_{1:N}|q}\big[\mathbb{P}(g(Y_1,...,Y_N) \notin \mathcal{X}(q) \mid X_{1:N}, q)\big]$$

$$= \mathbb{E}_{X_{1:N}|q}\left[\int_0^1 d\tau\ p(\tau)\ \mathbb{P}(g(Y_1,...,Y_N) \notin \mathcal{X}(q) \mid X_{1:N}, \tau)\right]$$

$$= \int_0^1 d\tau\ p(\tau)\ \underbrace{\mathbb{E}_{X_{1:N}|q}\left[\mathbb{P}(g(Y_1,...,Y_N) \notin \mathcal{X}(q) \mid X_{1:N}, \tau)\right]}_{=P_{\mathrm{err}}^{\mathrm{indep}}(N;q,\tau)}. \tag{20}$$

Therefore, $\lim_{N\to\infty} P_{\mathrm{err}}(N;q) = \lim_{N\to\infty}\int_0^1 d\tau\ p(\tau)P_{\mathrm{err}}^{\mathrm{indep}}(N;q,\tau)$. Since $P_{\mathrm{err}}^{\mathrm{indep}}(N;q,\tau) \in [0,1]$ for any $N \in \mathbb{N}$ and $\tau \in [0,1]$, we have that $p(\tau)P_{\mathrm{err}}^{\mathrm{indep}}(N;q,\tau) \in [0,p(\tau)]$, and therefore the integrand is bounded by $p(\tau)$, which integrates to 1. Therefore we can invoke the dominated convergence theorem and switch the limit and integral signs. Since by assumption $\lim_{N\to\infty} P_{\mathrm{err}}^{\mathrm{indep}}(N;q,\tau) = 0$ pointwise for any $\tau \in (0,1)$, we obtain $\lim_{N\to\infty} P_{\mathrm{err}}(N;q) = \int_0^1 d\tau\ p(\tau)\lim_{N\to\infty} P_{\mathrm{err}}^{\mathrm{indep}}(N;q,\tau) = 0$.

## B  Experimental Details

### B.1 Text-to-code generation

**Licenses.**  We use DeepSeek-Coder 7B (Guo et al., 2024), which is available under a permissive license that allows for both research and unrestricted commercial use. We report results on the MBPP dataset (Austin et al., 2021; Liu et al., 2023), released under an Apache license.

**Prompt template.**  We generate hypotheses with DeepSeek-Coder 7B (Guo et al., 2024) using the following prompt template:

> You are an AI programming assistant, utilizing the DeepSeek Coder model, developed by DeepSeek Company, and you only answer questions related to computer science. For politically sensitive questions, security and privacy issues, and other non-computer science questions, you will refuse to answer.
> ### Instruction:
> Please complete the following Python function in a markdown style code block:
> '''python
> [prompt]
> '''
>
> ### Response:
> '''python

**MBR-exec.** We use MBR-exec, an approach proposed by Shi et al. (2022) that consists of *(1)* sampling programs from an LLM, *(2)* executing each program on one test case, and *(3)* selecting the example with the minimal execution result-based Bayes risk. We use a 0/1 matching loss between execution results, and the Bayes risk of a program is defined by the sum of the loss between itself and the other sampled programs (the ground-truth program output is not used). We break ties by selecting the program with the smallest sampling index, corresponding to a random selection. See Shi et al. (2022, Section 3) for more details.

## B.2 Machine translation

**Licenses.** We use TowerInstruct 13B (Alves et al., 2024), which is released under a CC-BY-NC-4.0 license. We report results on the TICO-19 dataset (Anastasopoulos et al., 2020), publicly available through a Creative Commons CC0 license.

**Prompt template.** We generate hypotheses with TowerInstruct 13B (Alves et al., 2024) using the following prompt template:

```
<|im_start|>user
Translate the following [source language] source text to [target
language]:
[source language]: [source sentence]
[target language]: <|im_end|>
<|im_start|>assistant
```

**Visualizations.** In §5.2 we obtained a single reranking law for the all language pairs; we now fit different models for each language pair. Fig. 7 shows the log failure rate on the dev and test sets as a function of $N$ for EN-PT, EN-ES, and EN-RU. While the fits on the dev set are good, there is some degradation on the test set, especially for EN-ES (oracle and MBR decoding), possibly due to a shift in the distribution of scores/errors. We leave the investigation of more robust techniques and how to adapt to these cases for future work.

## B.3 Mathematical and commonsense reasoning

Our approach is fully general and can be useful in other domains other than code and language generation. In this subsection, we present additional experiments on mathematical and commonsense reasoning benchmarks, as prior work has shown that generating multiple hypotheses as an intermediate step is also advantageous in these scenarios (Wang et al., 2023).

We use samples generated by Aggarwal et al. (2023) with code-davinci-002, a GPT-3-based model with 175 billion parameters (Brown et al., 2020) which is part of the Codex series (Chen et al., 2021) (please refer to their Section 4 for more details; these samples were made publicly available by the authors at https://github.com/Pranjal2041/AdaptiveConsistency). We apply self-consistency over diverse reasoning paths (Wang et al., 2023), selecting the most frequent answer in the candidate set, and report results on the SVAMP (Patel et al., 2021) and StrategyQA (Geva et al., 2021) datasets. Following §5.1, we split the datasets in two equally sized parts to get development and test splits.

Similarly to Fig. 4, Fig. 8 shows the log failure rate on the dev and test sets (left and right, respectively) as a function of $N$, confirming that the same trends hold also for these two additional tasks.

## B.4 Computing infrastructure

Our insfrastructure consists of 2 machines, each equipped with 8 NVIDIA RTX A6000 GPUs (46GB) and 12 Intel Xeon Gold 6348 CPUs (2.60GHz, 1TB RAM). The machines were used interchangeably, and all experiments were executed on a single GPU.

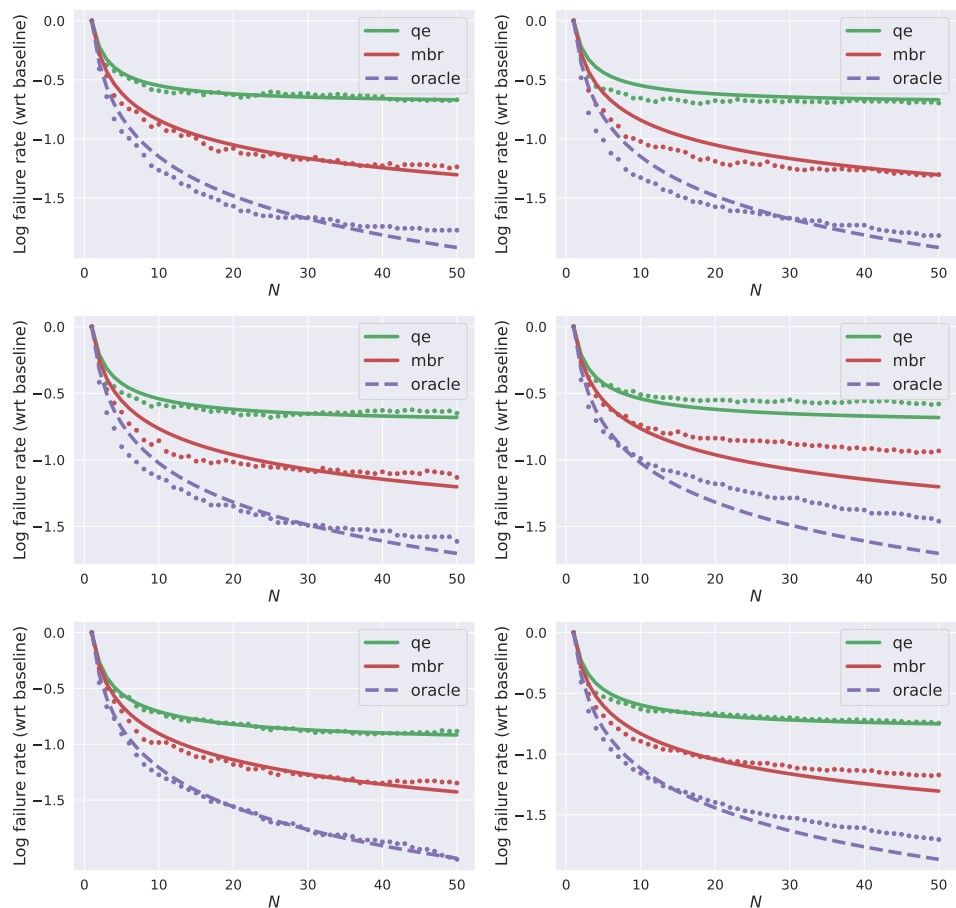

Figure 7: Log of the failure rate as a function of $N$. The empirical data is represented with dots (**left:** dev, **right:** test set) and our fitted models with solid and dashed lines (imperfect and perfect reranker, respectively). In this case, we fit separate models for each language pair (**from top to bottom:** EN-PT, EN-ES, and EN-RU).

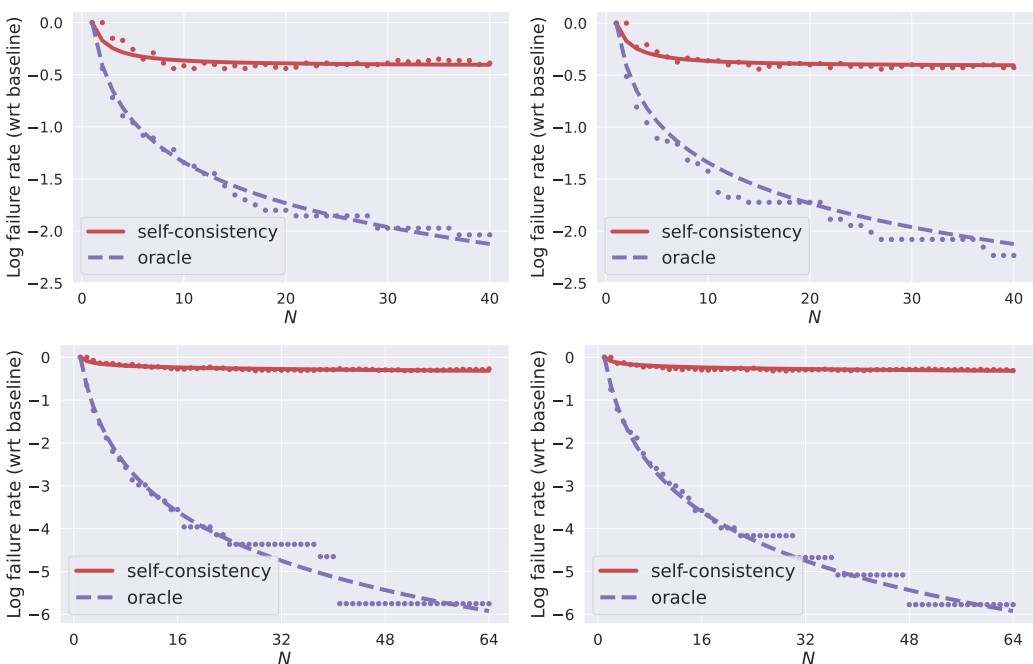

Figure 8: Log of the failure rate as a function of $N$. The empirical data is represented with dots (**left:** dev, **right:** test set) and our fitted models with solid and dashed lines (imperfect and perfect reranker, respectively). **Top:** mathematical reasoning on SVAMP . **Bottom:** commonsense reasoning on StrategyQA.

