# OpenReview forum: "Reranking Laws for Language Generation: A Communication-Theoretic Perspective"
_NeurIPS.cc/2024/Conference — NeurIPS 2024 spotlight_

### Official Review · Reviewer_8my9 · 2024-07-10

**Soundness:** 3
**Presentation:** 3
**Contribution:** 3
**Rating:** 5
**Confidence:** 4

**Summary:**

This paper proposes a reranking principle for language generation from a communication-theoretic perspective.  The paper conceptualizes the generator as a sender transmitting multiple descriptions of a message through parallel noisy channels. A receiver is designed to decode the message by ranking the descriptions and selecting the one found to be most reliable. Experiments show the effectiveness of proposed method in text-to-code generation task and machine translation of medical data task.

**Strengths:**

1 This paper proposes a reranking principle for language generation from a communication-theoretic perspective. The motivation is interesting and the theoretic analysis seems reasonable.

2 The paper is well-written and easily readable.

**Weaknesses:**

1 The related work analysis is not comprehensive. There are several ranking and reranking works in recommended systems, and none of them is mentioned or compared in this paper.

2 The experiments are not convincing enough. This paper only conducts two downstream experiments, i.e., Code generation and Machine translation. The results should be evaluated through more common and popular downstream tasks, such as QA (question choice) scenarios.

3 There is only 1 baseline for the text-to-code generation task and 2 baselines for the machine translation task. In particular, the one baseline for the text-to-code generation task is majority voting, which is not representative enough.

**Questions:**

Please see the weakness.

**Limitations:**

Yes

---

> ### Author Rebuttal · Authors · 2024-08-06
>
> Thank you for your review and suggestions. We are happy that you found our paper well written and the motivation and theoretical analysis interesting. We understand that your main concerns about our paper are related to our empirical validation—we address them below. We hope that this clarifies and alleviates your concerns.
>
> > “The related work analysis is not comprehensive. There are several ranking and reranking works in recommended systems, and none of them is mentioned or compared in this paper.”
>
> We welcome any suggestions you may have about related literature in reranking on recommender systems. Note, however, that the focus of our paper is very different: we are primarily concerned with reranking outputs of LLM generators and we study how the quality of the combined system is affected by the number of generated hypotheses and what their asymptotic properties are. Most of the works we are familiar with in the context of recommender systems study the different problem of, given a list of $N$ recommendations, reduce its size to $K<N$ elements through reranking.
>
> > “The experiments are not convincing enough. This paper only conducts two downstream experiments, i.e., Code generation and Machine translation. The results should be evaluated through more common and popular downstream tasks, such as QA (question choice) scenarios.”
>
> Thank you for the suggestion. We have run additional experiments on mathematical and commonsense reasoning benchmarks, and we observed that the same trends hold also for these two tasks, validating our method on other domains. We hope that this alleviates your concerns regarding the experimental part of our paper. Please see the general response for more details.
>
> > “There is only 1 baseline for the text-to-code generation task and 2 baselines for the machine translation task. In particular, the one baseline for the text-to-code generation task is majority voting, which is not representative enough.”
>
> Could you please clarify what you mean by “baseline”? Please note that our main goal is not to compare any method to a specific baseline but rather to validate our theoretical analysis using perfect and imperfect rerankers. Besides, we want to highlight that majority voting can be seen as a particular case of MBR decoding (Bertsch et al., 2023). In our experiments on text-to-code generation, we use MBR-exec (Shi et al., 2022), which is based on execution match. We would like to note that reranking methods relying on execution-based metrics are widely used in code generation (see, e.g., Chen et al., 2023; To et al., 2024). To clarify, as explained to Reviewer aDZG, MBR-exec consists of (1) sampling programs from an LLM, (2) executing each program on one test case, and (3) selecting the example with the minimal execution result-based Bayes risk. We use a 0/1 matching loss between execution results and the Bayes risk of a program is defined by the sum of the loss between itself and the other sampled programs. Since we are comparing the execution result of different programs, the Bayes risk will be minimal for the programs whose execution result is more frequent, hence the term “majority voting”. However, we understand that this term may be a bit misleading in this context and will update the paper accordingly. We will also use the additional page to include information on how MBR decoding works, instead of simply pointing to the papers.
>
> References:
>
> Chen et al., 2023. CodeT: Code Generation with Generated Tests.
>
> To et al., 2024. Functional Overlap Reranking for Neural Code Generation.

---

> > ### Comment · Reviewer_8my9 · 2024-08-13
> >
> > Thanks for your detailed response!
> >
> > As my concerns are eliminated, I will raise my score.

---

### Official Review · Reviewer_aDZG · 2024-07-12

**Soundness:** 3
**Presentation:** 3
**Contribution:** 3
**Rating:** 7
**Confidence:** 3

**Summary:**

A number of recent works in language generation can be framed as proposing two step methods, with a method to generate proposal strings, and another to rank these strings before choosing the best one to be output (this includes, e.g., MBR decoding).
This paper analyses this practice with a communication-theoretic approach.
They first assume that, given a query $q$, the goal of decoding is to generate a string $y$ in a set $\mathcal{X}(q)$.
They then assume the generator’s output $p(y_{1:N} \mid q)$ can be decomposed as $p(x_{1:N} \mid q)p(y_{1:N} \mid x_{1:N})$, with $x_n \in\mathcal{X}(q)$ and $p(y_{1:N} \mid x_{1:N})$ representing some kind of noise perturbation.
They then analyse the probability $p(rank(y_{1:N}) \notin\mathcal{X}(q)  \mid q)$ under different assumptions, showing that for many this value goes to zero as $N$ goes to infinity.
Finally, they run two experiments showing their theoretically derived predictions $p(rank(y_{1:N}) \notin\mathcal{X}(q)  \mid q)$ seem to correlate with the empirical probability of decoding errors.

**Strengths:**

The paper is well written and in general easy to follow (although I think section 4 could use a bit more hand-holding).

The paper provides an interesting theoretical analysis to a widely popular text generation framework.

The paper then investigates whether these theoretical insights are reflected empirically in real decoding settings.

**Weaknesses:**

In general, I liked this paper. In my opinion however, its main weaknesses are:
* limited empirical evaluation, with only two tasks, one generator model, and two reranking methods (besides an oracle ranker).
* the evaluation also makes some (in my opinion) debatable claims. E.g., in line 236 the authors state “[...] the imperfect reranker with majority voting, which fits the data well, as shown by the red curve.”. However, analysing Fig 4 (top), I would argue that the solid lines do not capture the data behaviour that well. In fact, the model’s performance seems to be empirically close to convergence with N, but the solid lines go monotonically down. Maybe running this analysis for larger values of $N$ would show whether the data indeed fits the predictions (specially if the predicted power law would generalise to larger values of $N$ as fit in the current data, and without fitting it on the new results).

**Questions:**

In the code generation experiments, the paper says “we use only one test case for each problem (Shi et al., 2022), and select one candidate by taking a majority vote over the execution results, dismissing hypotheses that fail to execute on the test case.” If a single test case is used, how is majority voting performed exactly? More details here could be helpful.


As a minor suggestion: I found the use of a “communication theoretical” framing here a bit distracting, and it seems to me it could be discarded with no significant change to the paper’s contributions. The authors, for instance, discuss error correcting codes early in the paper, but then they (admittedly) do not require generated strings to be error-corrected. (The selected string simply needs to be in an acceptable set $rank(y_{1:N}) \notin\mathcal{X}(q) $.) Besides, the generator and ranker are framed as a sender and a receiver—with a noisy channel in between them—but no message is actually decoded by the receiving ranker. Alternatively, highlighting the role that a communication-theoretical framing has in the paper (and why it is needed) could be useful.

**Limitations:**

The authors properly discuss their analysis limitations in the paper.

---

> ### Author Rebuttal · Authors · 2024-08-06
>
> Thank you for your positive review and suggestions. We are glad that you found our paper well written and easy to follow, and the theoretical analysis interesting. We address below your main concerns.
>
> > “limited empirical evaluation, with only two tasks, one generator model, and two reranking methods (besides an oracle ranker).”
>
> Thanks for pointing this out. While our focus was on machine translation and text-to-code generation, we have run additional experiments on mathematical and commonsense reasoning benchmarks and observed that the same trends hold also for these two tasks, validating our method on other domains. We hope that this alleviates your concerns regarding the limited empirical evaluation of our method. Please see the general response for more details.
>
> > “the evaluation also makes some (in my opinion) debatable claims (...) the authors state “[...] the imperfect reranker with majority voting, which fits the data well (...) analysing Fig 4 (top), I would argue that the solid lines do not capture the data behaviour that well (...) Maybe running this analysis for larger values of N would show whether the data indeed fits the predictions (...).”
>
> As discussed in the limitations section, while the experiments suggest a reasonable fit, you are right that the fit is not perfect and we will adjust the text accordingly. In fact, for large $N$, errors are rare events, and therefore prone to statistical inaccuracies (this is visible in the “steps” observed in the code generation plots). For text-to-code generation, in practice, most work does not use more than 200 samples due to the increased cost. For LLM-based machine translation, the work of Farinhas et al. (2023), which we use in our experiments in Section 5.2, suggests that using a smaller $N$ is enough. Even though this is not discussed in the paper, the cost of MBR decoding grows quadratically with the number of hypotheses $N$, making it impractical to try values higher than these ones. In any case, both results appear to be consistent with the new tasks we experimented on (as mentioned in the previous point).
>
> > “In the code generation experiments, the paper says “we use only one test case for each problem (Shi et al., 2022), and select one candidate by taking a majority vote over the execution results, dismissing hypotheses that fail to execute on the test case.” If a single test case is used, how is majority voting performed exactly? More details here could be helpful.”
>
> We agree more details will be helpful – due to space constraints we ended up trimming some details about specific reranking techniques such as MBR decoding or reranking based on quality estimation. In this particular experiment, we follow MBR-exec, an approach proposed by Shi et al. (2022) that consists of (1) sampling programs from an LLM, (2) executing each program on one test case, and (3) selecting the example with the minimal execution result-based Bayes risk. We use a 0/1 matching loss between execution results and the Bayes risk of a program is defined by the sum of the loss between itself and the other sampled programs (of course, the ground-truth program output is not used). This is described in detail in the original paper (see., e.g., their Section 3). We break ties by selecting the program with the smallest sampling index, corresponding to a random selection. For completeness, for machine translation we followed the exact same procedure as described in Farinhas et al. (2023). In this case, as described in L243-246, MBR decoding does not use “execution results” but is rather based on a utility function based on a reference-based metric (in our case, Comet-22). We agree that this information should be described in more detail, and we will add descriptions of all the methods that we used for text-to-code generation and machine translation in a dedicated section.
>
> Thank you for the suggestions!

---

> ### Comment · Reviewer_aDZG · 2024-08-14
> **Response to Authors**
>
> I thank the authors for their response. I still think that running this analysis for larger values of $N$ would be good (or improving how the paper assesses that the data fits its predictions), but I have increased my scores due to the extra experiments added. I think this is an interesting paper that should be accepted.

---

### Official Review · Reviewer_o8DD · 2024-07-12

**Soundness:** 4
**Presentation:** 4
**Contribution:** 3
**Rating:** 8
**Confidence:** 4

**Summary:**

This paper proposes to regard generator-reranker LLMs, i.e., LLMs generating multiple outputs and then reranking them, as communication systems. The idea is to consider the outputs noisy with the objective for the reranker to find the less noisy one.

**Strengths:**

- the approach is very flexible. It doesn’t depend on a particular architecture and the outputs to rerank can be generated by multiple different models.
- a sound parallel is made with communication theory which helps to understand why this approach works.
- the approach is well formalized. I couldn’t find any error but be aware that I’m not very familiar with Zipf-Mandelbrot and Mallows model.
- Two scenarios are taken into account: with and without an independence assumption
- Experiments with machine translation are very relevant for generator-reranker LLMs

**Weaknesses:**

- Absence of analysis of the experiment results. This is a critical weakness of the paper. The experimental settings are described, and the results are given (plots), but without any comment. For the MT experiments, the authors wrote that they got some scores, and then we have the next section.

**Questions:**

Please comment on your results. What do you conclude from them? Why are they insightful (or not insightful)? How can they be used in future work? etc.
The paper is really good but we miss some analysis.

**Limitations:**

Limitations are adequately addressed.

---

> ### Author Rebuttal · Authors · 2024-08-06
>
> Thank you for your positive review and suggestions. We are happy that you found our approach to be well formalized and flexible, the parallel with communication theory useful, and the experiments relevant.
>
> We agree that the paper would benefit from more discussion about how our results can be  useful in practice. Sections 6 and 8 already provide some information about this (e.g., the reranking laws allow us to predict how many hypotheses are necessary to achieve a desired error probability), but we will update the manuscript with a more specific analysis in Section 5. Additionally, we have run additional experiments on mathematical and commonsense reasoning benchmarks. Similar trends hold for the new experiments, which confirms the general applicability of our approach and further validates our theoretical model.  Please see the general response for more details.

---

### Official Review · Reviewer_EFsu · 2024-07-12

**Soundness:** 4
**Presentation:** 4
**Contribution:** 3
**Rating:** 7
**Confidence:** 4

**Summary:**

The paper provides a framework for understanding the theoretical properties of generator-reranker systems for language generation. It relates the reranking process to error correction during the decoding of messages in noisy channels, a concept that has been well-studied in communication theory. Explicitly, the paper conceptualizes an LLM generator as a sender transmitting messages through noisy channels, with the reranker acting as the receiver decoding these messages. This framing explains why 1) things like redundancy in the set of generated strings are helpful in generator-reranker systems, actually increasing the likelihood of an acceptable output and 2) increasing the number of options from the generator in the reranking process generally increases system performance. The paper makes several theoretical contributions, showing that when generator-reranker systems meet certain theoretical requirements, there is a guarantee of “error-free” performance. This property holds even when channel distributions are not independent, i.e., when the same model is used to generate possible solutions to the input. The paper provides some empirical verification of their proposed laws.

**Strengths:**

The paper is very well written. The math is clearly explained and sound; it provides a nice theoretical justification of why generator-reranker systems work well  The topic is also very relevant, since LLM reliability (which is improved by the generator-reranker paradigm) is of utmost concern. The laws proposed in this paper also have practical use: they would allow practitioners to decide the number of strings needed from the generator system to achieve a certain accuracy, without lots of trial and error.

**Weaknesses:**

The applicability of the communication system framework to generator-reranker systems is somewhat questionable given that there is not the same binary notion of acceptable/unacceptable for language generation systems. Rather, we’re dealing with a continuous spectrum of quality and the appeal of the generator-reranker system is its ability to increase quality (perhaps amongst “acceptable” solutions) rather than move from the realm of unacceptable to acceptable answers. The impact of this difference between the theoretical framework and the evaluation of generation systems in practice isn’t really discussed.
There are a few points that could be addressed to improve readability:
* Some aspects of the abstract/intro are confusing because terms have not been defined and their equivalences in an LLM generator-reranker system have not yet been specified. For example, the reader won’t know what the implications of “channel distributions being statistically dependent” are (mentioned in the abstract) until after they’ve read through much of the paper.
* There isn’t much intuition about what the R.V. X corresponds to in the generator-reranker system
* Some more intuition behind the scale parameter (other than just the settings that it gives for its extreme values) would be helpful


The computational experiments are not very comprehensive, exploring only two generation tasks (one generator model for each task). It is thus unclear how general their results are.

**Questions:**

* I didn't understand the (bolded) comment in lines 148-9. It makes it sound as though the quality of the reranker depends on the quality of the generator. Could this be clarified?
* Minor style recommendation: Perhaps move the first sentence of 3.3 to right after providing the expression for the partition function. That feels like a more natural place to me. In footnote 1: equivalent class -> equivalence class

**Limitations:**

The authors discuss most of the limitations present in their work. I would like to see an additional discussion of the implications of their results for continuous evaluation metrics (rather than binary acceptable/unacceptable)

---

> ### Author Rebuttal · Authors · 2024-08-06
>
> Thank you for your positive review and suggestions. We are glad that you found our paper to be very well written, the math to be clear and sound, and our method to have practical use. We address below your concerns about our paper.
>
> > “there is not the same binary notion of acceptable/unacceptable (...) I would like to see an additional discussion (...) for continuous evaluation metrics”
>
> This is a very good suggestion. Our framework can indeed be extended to continuous evaluation metrics, although some concepts (e.g. the notion of “asymptotically error-free”) would need to be modified accordingly. We sketch below some ways in which this extension could be made:
> - We would need to posit a probability density for the continuous evaluation metric (instead of a Bernoulli error probability) for each hypothesis coming from the generator. In the simplest case, this could be a Gaussian distribution with some input-dependent mean and variance. For bounded evaluation metrics (e.g. between 0 and 1) other families would be more suitable (e.g. uniform or Beta distributions).
> - For a perfect reranker and independent hypotheses, the resulting output after reranking would be distributed according to the corresponding **extreme value distribution** (this models the distribution of the highest quality score among the $N$ hypotheses). Extreme value distributions are an important subject of study in order statistics. For example, for the Gaussian case above, we would obtain a Gumbel distribution, for uniform we obtain Beta, etc. The asymptotic case ($N  \rightarrow \infty$) corresponds to one of Gumbel, Fréchet or Weibull families (this is a consequence of the Fisher–Tippett–Gnedenko theorem [1]).  From the extreme value distribution, we can obtain the expected quality score or the probability of a quality score being below a threshold.
> - Unfortunately, the generalization to imperfect rerankers (e.g. Mallows or Zipf-Mandelbrot rerankers) seems much harder than in the binary case.
>
> In the paper, we opted for focusing on the binary acceptable/unacceptable case for three main reasons: (1) This case is simpler to analyze and to understand (particularly when rerankers are imperfect). (2) It is still highly relevant in practice – e.g., in code generation, as well as other tasks, the code either executes and gives the correct answer, or it doesn’t (regardless of its quality). (3) It would be very hard to cover both the binary and continuous cases in the right level of detail in a single 9-page paper, hence we decided to go deeper on the former and leave the latter for future work.
>
> Yet, we will use the additional page to add this discussion, as suggested.
>
> [1] David, Herbert A.; Nagaraja, Haikady N. (2004). Order statistics. John Wiley & Sons. p. 299.
>
>
> > “Some aspects of the abstract/intro are confusing because terms have not been defined and their equivalences in an LLM generator-reranker system have not yet been specified.”
>
> Thank you for pointing this out. The paragraph in L41-47 and Figure 1 provide a brief explanation of how an LLM generator-reranker system can be seen as a communication system, but we agree that the wording might not be easily understood for a first-time reader. We will improve the text and update the caption of Figure 1 to include more information, hopefully making it more clear.
>
> > “There isn’t much intuition about what the R.V. X corresponds to in the generator-reranker system”.
>
> In our setup, we assume that a sender transmits $N$ message descriptions in parallel through noisy channels, resulting in the generation of $N$ potentially corrupted hypotheses $y_i, i\in\{1,...,N\}$. The LLM generator consists of both the sender and the noisy channels, where the $y_i, i\in\{1,...,N\}$ are the $N$ hypotheses sampled from the model. For example, in a machine translation scenario, these $y_i$ correspond to $N$ alternative translations, each potentially containing errors. The message descriptions $x_1, ..., x_N \in \mathcal{X}(q)^N$ correspond to acceptable answers within the equivalence class $\mathcal{X}(q)$ (as explained in footnote 1), before being corrupted by the noisy channel. This will be clarified in the final version.
>
> > “Some more intuition behind the scale parameter (...) would be helpful”
>
> We agree that this can be further clarified in the paper. We mentioned the cases of a random reranker ($e^{-\lambda}=1$) and a perfect reranker ($e^{-\lambda}=0$). For a Mallows model, $e^{-\lambda}$ strictly between 0 and 1 correspond to imperfect rerankers that are better than random. The lower this value, the higher the quality of the reranker. Thus, $e^{-\lambda}$ works as an inverse measure of reranker quality. We will clarify.
>
> > “The computational experiments are not very comprehensive (...).”
>
> We agree that the paper becomes stronger if we report experiments in more tasks beyond machine translation and text-to-code generation. We have run additional experiments on mathematical and commonsense reasoning benchmarks, validating our method on other domains. Similar trends hold for the new experiments, which confirms the general applicability of our approach. We hope that this alleviates your concerns. Please see the general response and attached figure for details.
>
> (_continues in a follow-up comment_)

---

> ### Author Response · Authors · 2024-08-06
>
> > “I didn't understand (...) lines 148-9 (...) the quality of the reranker depends on the quality of the generator.”
>
> Thank you for letting us know that you found this part unclear. We want to clarify that the quality of the reranker itself is independent of the quality of the generator. While both perfect and Mallows rerankers achieve exponentially decreasing error probabilities as the number of hypotheses $N$ increases, the exact rate of convergence is different. For the Mallows reranker, the rate of convergence also depends on the parameters of the Mallows model ($\lambda$). As shown by Eq. (2), $P_\mathrm{err}(N; q)$ decays exponentially as $\epsilon^N$, where $\epsilon$ is the probability of generating an unacceptable hypothesis. For a Mallows model, Proposition 1 shows that the error probability also decays exponentially, with $P_\mathrm{err}(N; q) = \mathcal{O}((e^{-\lambda}(1-\epsilon) + \epsilon)^N)$. While the convergence rate $e^{-\lambda}(1-\epsilon) + \epsilon$ is generally greater than $\epsilon$, it still ensures an exponentially decreasing error probability as $N$ increases. Thus, what we meant with the bolded comment in lines 148-9 is that Mallows rerankers behave asymptotically like perfect rerankers but with a higher effective error probability, due to the additional factor depending on $e^{-\lambda}$. That is, asymptotically, a bad (Mallows) reranker is equivalent to a perfect reranker with a worse generator. We will clarify this in the final version.
>
> > “move the first sentence of 3.3 to right after providing the expression for the partition function.”
>
> Even though this result is not used in the previous section, we agree with the recommendation. We will update the paper accordingly, keeping a reminder in Section 3.3.
>
> > “In footnote 1: equivalent class -> equivalence class.”
>
> Thanks for pointing this out, we will fix the typo.

---

### Author Rebuttal · Authors · 2024-08-06

Dear reviewers,

We appreciate the time and effort you have taken to review our paper and provide constructive feedback. We are pleased to see that our work has been positively received.

The main weakness pointed out by the reviewers is that we illustrate our reranking laws on two applications only, code generation and machine translation. We picked these tasks due to their particularly challenging nature and the prevalent use of reranking techniques to improve model performance. However, we believe our approach is fully general and can be useful in other domains as well. Therefore, we followed the reviewers' suggestions and ran additional experiments on mathematical and commonsense reasoning benchmarks, as prior work (Wang et al., 2023) has shown that generating multiple hypotheses as an intermediate step is also advantageous in these scenarios. We used samples generated by Aggarwal et al. (2023) with code-davinci-002, a GPT-3-based model with 175 billion parameters (please refer to their Section 4 for more details; these samples were made publicly available by the authors at https://github.com/Pranjal2041/AdaptiveConsistency). We applied self-consistency over diverse reasoning paths (Wang et al., 2023),  selecting the most frequent answer in the candidate set. We report results on SVAMP (Patel et al., 2021) and StrategyQA (Geva et al., 2021). The attached pdf includes plots similar to Figure 4 in our manuscript, showing the log failure rate as a function of N. We observed that the same trends hold also for these two additional tasks. We hope this alleviates the main concern raised by the reviewers.

References:

Wang et al., 2023. Self-Consistency Improves Chain of Thought Reasoning in Language Models.

Aggarwal et al., 2023. Let’s Sample Step by Step: Adaptive-Consistency for Efficient Reasoning and Coding with LLMs.

Patel et al., 2021. Are NLP Models really able to Solve Simple Math Word Problems?

Geva et al., 2021. Did Aristotle Use a Laptop? A Question Answering Benchmark with Implicit Reasoning Strategies.

---

### Decision · Program_Chairs · 2024-09-25

**Decision:**

Accept (spotlight)

**Comment:**

This paper presents an understanding of generator-reranker systems from a communication-theoretic perspective.  The paper's main claim is that these systems can be asymptotically error free as more and more samples are drawn, even if the ranker is noisy. This is analyzed both theoretically and empirically in two domains, with additional results included during the rebuttal period.

The reviewers mostly praised the core ideas of this paper, its writing, and its theoretical contribution. EFsu gives some good suggestions for improvement. I agree with Efsu that there is a somewhat major conceptual weakness of the paper in that it treats generation in a binary correct/not correct fashion. However, the authors' rebuttal is quite informative here and will be a good addition to the paper.

Overall, the interestingness of the paper outweighs any concerns about lack of more extensive empirical evaluation or development.